# Neural Correlates of Mobility in Children with Cerebral Palsy: A Systematic Review

**DOI:** 10.3390/ijerph21081039

**Published:** 2024-08-07

**Authors:** Isabella Pessóta Sudati, Diane Damiano, Gabriela Rovai, Ana Carolina de Campos

**Affiliations:** 1Department of Physical Therapy, Child Development Analysis Laboratory (LADI), Federal University of São Carlos (UFSCar), São Carlos 13565-905, SP, Brazil; grovai@estudante.ufscar.br; 2Rehabilitation Medicine Department, Clinical Center, National Institutes of Health (NIH), Bethesda, MD 20892, USA; damianod@cc.nih.gov

**Keywords:** cerebral palsy, children, brain mapping, cerebral activity, mobility

## Abstract

Recent advances in brain mapping tools have enabled the study of brain activity during functional tasks, revealing neuroplasticity after early brain injuries and resulting from rehabilitation. Understanding the neural correlates of mobility limitations is crucial for treating individuals with cerebral palsy (CP). The aim is to summarize the neural correlates of mobility in children with CP and to describe the brain mapping methods that have been utilized in the existing literature. This systematic review was conducted based on PRISMA guidelines and was registered on PROSPERO (n° CRD42021240296). The literature search was conducted in the PubMed and Embase databases. Observational studies involving participants with CP, with a mean age of up to 18 years, that utilized brain mapping techniques and correlated these with mobility outcomes were included. The results were analyzed in terms of sample characteristics, brain mapping methods, mobility measures, and main results. The risk of bias was evaluated using a checklist previously created by our research group, based on STROBE guidelines, the Cochrane Handbook, and the Critical Appraisal Skills Programme (CASP). A total of 15 studies comprising 313 children with CP and 229 with typical development using both static and mobile techniques met the inclusion criteria. The studies indicate that children”with’CP have increased cerebral activity and higher variability in brain reorganization during mobility activities, such as gait, quiet standing, cycling, and gross motor tasks when compared with children with typical development. Altered brain activity and reorganization underline the importance of conducting more studies to investigate the neural correlates during mobility activities in children with CP. Such information could guide neurorehabilitation strategies targeting brain neuroplasticity for functional gains.

## 1. Introduction

Cerebral palsy (CP) is characterized by activity limitations caused by a group of disorders of movement and posture due to an injury in the developing brain [1]. Mobility, defined by the ability to move around in the environment to participate in daily activities and interact with family and society [2], is often limited in individuals with CP. In this study, we focused on the mobility activities outlined in Chapter 4 of the Activity and Participation section of the International Classification of Functioning, Disability and Health (ICF) [3].

Several body functions and structures may contribute to mobility limitations in CP, including spasticity, range of motion, and altered selective motor control, and have been shown to impact gait pattern [4]. Changes in spatiotemporal (e.g., reduced gait velocity) and kinematic variables have been extensively described [5] and have been shown to correlate with mobility level [6].

The significance of motor findings in CP also may be influenced by injury-related factors such as location, timing, and extent [7,8]. White matter lesions are the most common brain injuries that occur in children with CP and are often associated with mild gross motor dysfunction [9]. The maturation of corticospinal connections typically commences around the last semester of pregnancy and may continue into the first two years of life [10]. In CP, damage in corticospinal pathways at critical periods of central nervous system maturation is also correlated with motor disturbances [8,11].

The immature central nervous system is believed to have greater potential for neuroplasticity [12]. The developing brain that has structural damage has the capacity for change as with all infants but may develop different patterns of cortical organization in response to altered motor experiences. This may take place through processes such as the involution of damaged cortical projections, greater retention of ipsilateral projections, or a shift in hemispheric control to accommodate for regions that were injured, which has been well demonstrated in studies of the upper limb function after unilateral brain injury [10,13]. Although it is acknowledged that the neuroplastic events occurring after early brain injuries impact the functional organization of the brain, their consequences on mobility are not well understood. Functional rearrangement of the brain following early brain injuries is complex and may not be fully captured by structural imaging.

For a long time, magnetic resonance imaging (MRI) has been the sole brain imaging technology most used to describe brain injuries in individuals with CP and their motor correlates [14]. However, this technique has limitations in terms of describing the complex dynamics of brain functioning, especially during more complex or functional motor tasks. Many children with CP have difficulty remaining still in the scanner due to involuntary movement or exaggerated startle responses, which may be more exacerbated if asked to perform a movement, which warrants additional brain mapping approaches.

Brain mapping techniques can be divided into two groups, based on their principles: electrophysiological and hemodynamic analysis [15]. In the former, it is possible to study the electrocortical activity through the apical dendrites of large pyramidal neurons (e.g., electroencephalogram (EEG); magnetoencephalogram (MEG); transcranial magnetic stimulation (TMS, which does not directly perform brain mapping but in combination with other techniques allows functional localization of the underlying cortical areas), whereas the latter identifies changes in oxyhemoglobin and deoxyhemoglobin concentrations related to neural activity (e.g., functional magnetic resonance imaging (fMRI); functional near-infrared spectroscopy (fNIRS)) [15].

There has been growing interest in understanding brain behavior during functional activities, which has become possible due to the development of non-invasive technologies, particularly those that are more tolerant to motion. Both EEG and fNIRS have arisen as promising alternatives to evaluate brain activity while performing dynamic tasks in more naturalistic settings. While mainly confined to the cortical regions and with lower spatial resolution, these techniques allow the identification of cerebral demands during mobility tasks, with applications such as establishing treatment goals for focal stimulation of neuroplasticity possible [16,17].

With EEG, electrophysiological cortical activity can be reported as alpha (mu), beta, and gamma band, defined as brain waves in the frequency range of 8–13 Hz, often associated with relaxed, wakeful states; brain waves in the frequency range of 14–30 Hz, associated with active thinking, focus, and movement preparation; and high-frequency brain waves ranging from 30 to 100 Hz, linked to high-level cognitive functions, such as perception, problem-solving, and consciousness, respectively [18]. Other relevant outcomes include task-related desynchronization (TRD), defined as a reduction in the power of specific frequency bands during a cognitive or motor task, indicating increased neuronal activity; and task-related synchronization (TRS), defined as an increase in the power of specific frequency bands during or after a task, indicating increased synchronous neuronal activity, often associated with the completion of a task or a state of relaxation [18]. Central-Frontal Coherence is a measure of the functional connectivity between central and frontal regions of the brain, indicating how well these areas communicate during cognitive and motor tasks. Altogether, these outcomes can support the understanding of neural correlates of mobility performance.

fNIRS outcomes such as peak and total changes in oxy- (HbO), deoxy- (HbR), and total hemoglobin can reliably identify underlying cortical areas that are active during motor tasks and are a promising way to explore the effects of neurorehabilitation [19]; other outcomes such as resting state connectivity have also been reported to reveal mechanisms associated with motor ability in children with neurological disorders [20].

Despite the potential of these techniques, previous reviews have only explored the correlates of structural imaging with motor function in CP [10,21] and were limited to correlations of motor classification and motor type with brain findings [21]. Other reviews addressing fNIRS mapping in children were not focused on CP or mobility outcomes [22,23]. There is therefore a lack of systematic reviews exploring functional imaging and mobile brain mapping techniques that allow cortical mapping during functional and dynamic mobility tasks in children with CP. However, challenges related to brain mapping technologies may have contributed to a greater focus on children with higher levels of functioning (those with unilateral CP in GMFCS Levels I and II) and on upper limb function, even in studies including children and/or individuals with mobility limitations [9,21].

Given the importance of understanding brain behavior during mobility tasks in this population, this systematic review may contribute to summarizing findings related to brain activation during mobility tasks. In addition, understanding the cerebral adaptations during these tasks may support the planning and execution of interventions targeting altered activation patterns to potentially maximize neuroplasticity and the functional impact of treatments. The aim of this systematic review was to describe the methods utilized and their results in an effort to summarize what is now known about the neural correlates of mobility in children with CP.

## 2. Materials and Methods

### 2.1. Design and Ethical Approval

This systematic review was conducted according to PRISMA guidelines [24] and registered in PROSPERO (n° CRD42021240296).

### 2.2. Search Strategy

The literature search was conducted in April 2023 from the PubMed/Medline and Embase (Elsevier) databases. Search updates were performed every month. The following keywords and their synonyms were used in the string search “cerebral palsy”, “child”, “mobility” and “cerebral activity”.

The search terms and strategy were adapted according to each database (Appendix A). A manual search by reviewers was performed based on the reference lists of studies initially found.

### 2.3. Study Selection

Inclusion criteria were the following:Participants diagnosed with CP (at least 50% of the CP group sample);Mean sample age up to 18 years;Use of brain imaging or mapping techniques (fMRI, EEG, fNIRS, MEG, among others);Correlation with mobility measures (e.g., measures of locomotion, postural transfers, gross motor function),Observational study design (or intervention studies if baseline data are available).

Studies were excluded if the participants had other health conditions than CP (e.g., traumatic brain injury, genetic disorders), if no measures of mobility were used (e.g., focus on upper limb activities), and if the publication type was other than a peer-reviewed research paper (e.g., conference abstracts, dissertations/theses, commentaries, or systematic reviews). There were no time and language restrictions. Two reviewers (IPS and ACC) independently performed the selection of studies based on the titles or abstracts of all studies identified by the initial search. In case of disagreement, a consensus was reached. Afterwards, the reviewers independently extracted the data and rated the quality of included studies.

### 2.4. Methodological Quality Assessment

The risk of bias was evaluated by a checklist created and used previously by our research group [25,26] that is based on the guidelines from Strengthening the Reporting of Observational Studies in Epidemiology (STROBE) from the literature on the development of quality criteria described in the Cochrane Handbook for Systematic Reviews, and from the Critical Appraisal Skills Programme (CASP). The checklist considered the following aspects: (1) presentation of study objectives; (2) rationale for study hypotheses; (3) use of appropriate design to meet objectives; (4) participant delineation, (5) inclusion criteria proposed by the study; (6) exposition of volunteer recruitment; (7) description of sampling type; (8) ethical aspects; (9) volunteers not participating in or excluded from the study; (10) sample computation for volunteer selection; (11) description of variables; (12) use of appropriate statistical methods to analyze the result; (13) descriptive measures of precision or variability of study results; (14) the study’s external validity; (15) findings in a clear, objective manner; and (16) the study’s limitations [25,26]. Each item was scored as 1 if the study met the requirement and 0 if it did not address it. Scores between 12 and 16 points are considered good quality; scores between 7 and 11 points as fair; and scores less than 7 points as poor quality.

### 2.5. Data Extraction and Analysis

A descriptive analysis was used to report the following extracted data: authors, participants, age, sex, CP type, Gross Motor Function Classification System (GMFCS) level, type of brain imaging technology, device parameters, variables analyzed, mobility measures, and main results.

## 3. Results

### 3.1. Study Selection

A total of 2445 studies were retrieved, 47 being identified as duplicates. A total of 66 were selected for full-text review. From these, studies were excluded if they did not have mobility outcomes (*n* = 20) or were conference studies (*n* = 18) among other reasons described in detail in Figure 1.

In total, 13 studies met the inclusion criteria. A manual search resulted in two additional studies being included. Therefore, 15 studies were reviewed.

### 3.2. Participants and Study Characteristics

The characteristics of the study samples are listed in Table 1. The 15 included studies comprised a total of 542 subjects, including 313 children with CP and 229 with typical development (TD). Sample size ranged from 8 to 65 participants (median = 28), with a mean age of 12.1 ± 3 years.

### 3.3. Main Results

The main associations of brain mapping with mobility measures in each of the included studies are shown in Table 2. The most frequently associated measure was kinematic analysis [29,30,31,34,35,36,37,38,39,40]. Some studies used clinical scales such as the Pediatric Evaluation of Disability Inventory Computer Adaptive Test (PEDI-CAT) [28], AbiLOCO [41], Gross Motor Function Measure-66 (GMFM-66) [33], GMFM-88 and Gait Profile Score [31,37], and functional tests [27]. Brain mapping technologies included diffusion tensor imaging (DTI) plus TMS (*n* = 1 study), DTI plus functional MRI (fMRI; *n* = 4 studies), EEG (*n* = 2 studies), fNIRS (*n* = 2 studies), MEG (*n* = 1 study), MEG plus MRI (*n* = 2 studies), resting-state fMRI (rs-fMRI; *n* = 1 study), and TMS (*n* = 2 studies). Only four studies performed the assessment of mobility concomitantly with brain mapping [30,34,39,41].

In general, children with CP have greater mu and beta event-related desynchronization (ERD), oxyHB concentration, and greater coherence in both hemispheres during gait when compared to children with TD [30,34,39]. Those on whom an MEP can be elicited have higher gait speed than those without MEPs [31]. In addition, during cycling, children with CP have a greater number of active channels when compared to their peers without CP [41].

Considering the simultaneous assessment of mobility and brain activity, increased cortical activation in individuals with cerebral palsy (CP) during mobility tasks suggests greater neural resource demands and is associated with poorer motor function and delayed cerebral maturation compared to typically developing (TD) individuals [30,34,39,41].

### 3.4. Methodological Quality Assessment

According to the checklist, 6 of the 15 studies were classified as having good methodological quality [28,29,30,31,32,33] (Table 3). All studies complied with items 1, 4, 8, 11, 12, 14, and 15. None reported the sampling method (item 7) or justified the sample size or measures of effect size (item 10).

## 4. Discussion

This study reviewed the existing literature about neural correlates of mobility in CP.

### 4.1. Study Characteristics

Fifteen studies were selected, comprising a total study population of 542 children, 313 of them diagnosed with CP. The majority of studies had a small sample size and tended to include young children, resulting in an average age of 12 years. These studies were likely samples of convenience that in some cases involved some mild discomfort from the stimulation or a tight-fitting cap and required participants to remain still for brief periods and to follow directions on when and how to move which may explain the largely school-age range common across studies.

Most studies included children with CP classified at the mildest levels of functional mobility (GMFCS levels I, II, and III) which limits the understanding of the cerebral activity of children with significant motor impairment. Most studies included children with both bilateral and unilateral CP, which may have different relationships between neural patterns and motor functioning. Although mobility limitations are a defining feature in children with CP and the focus of extensive interventions, there are few studies investigating cerebral activity during these tasks; however, the technical issues involved in brain imaging while walking can be challenging and are only possible with mobile technologies.

### 4.2. Assessment Methods

Mobility assessments

Various tools were used to assess mobility. Self-selected or comfortable walking speed was the outcome most used to evaluate mobility [27,29,30,31,34,35,36,37,38,39,40], followed by GMFM-88 or 66 [31,33].

Kinematic data from clinical gait analysis identify the angular excursions of different joints or segments [42] and are frequently used to describe gait patterns [43] at the level of body structures and functions. However, temporal-spatial measures such as gait speed in particular, collected during 3D gait analysis or recorded separately, are good indicators of mobility. Clinical gross motor or mobility scales are commonly used in studies with CP because they are low-cost, validated instruments [44]. The multiple measures utilized across studies, however, make it difficult to summarize results.

Brain-mapping techniques

Brain imaging techniques investigate the human brain function with non-invasive technologies that include electrophysiological tools, quantifying changes in electrical or electromagnetic activity, and hemodynamic tools, quantifying changes in metabolic activity [15]. In this study, brain mapping technologies included both static (DTI, MEG, fMRI, and TMS) and mobile techniques (EEG and fNIRS).

Static brain-mapping techniques usually provide qualitative and quantitative information about the brain structure. They have been largely used to establish the etiology of lesions [14], for functional localization of cortical areas [15], and diffusion-based measurements [45], and have demonstrated correlations with clinical findings [46]. Static techniques, however, are limited to rest conditions or single joint /small movements.

On the other hand, mobile techniques allow the study of brain activity during functional motor tasks, such as gait [47]. These portable technologies can be used in more naturalistic environments and, in most cases, do not require an expensive system to perform the assessments [48,49]. They also provide additional benefits in terms of assessing individuals who may not be able to stay sufficiently still during static procedures, such as children with motor disorders [50].

### 4.3. Neural Correlates of Mobility Tasks

Increased brain activity reflects altered mobility in CP

Although several similarities exist between brain activity in CP and TD, such as the presence of task-related desynchronization during treadmill walking [30,39], a variety of studies using different techniques suggest that a greater extent and/or magnitude of brain activation is often observed in CP than in TD during mobility tasks, with higher values tending to correlate with poorer mobility.

Using EEG, Short et al. (2020) demonstrated that treadmill gait resulted in higher cortical activation in both hemispheres in unilateral CP [39]. The increased demand for neural resources may extend beyond the alpha and beta bands as seen in TD, to increased gamma band activity in CP [39]. Additionally, George et al. (2020) reported larger central-frontal coherence in CP compared to controls, which illustrates greater use of cortical resources to maintain standing posture [30]. George et al. (2020) also reported lower and more variable peak frequencies in the alpha (mu) event-related power in CP in both hemispheres which may reflect delayed cerebral maturation, even though the participants presented mild motor involvement [30,51].

fNIRS studies provide further support for increased brain activity in CP through hemodynamic results related to gait [34] and cycling/hip flexion/dorsiflexion tasks [41]. During gait, greater variability in the gait temporal-spatial data was correlated to greater activity in sensorimotor cortices of children with CP, again suggesting that increased utilization of cortical resources is associated with poorer gait function [34]. In the fNIRS investigation of isolated ankle and entire lower limb tasks by Sukal-Moulton et al. (2018), a higher extent and magnitude of total hemoglobin in CP were directly proportional to the degree of motor involvement: those in GMFCS level III had significantly higher activation in the sensorimotor cortex compared to GMFCS I-II and TD [41].

Cortical overflow also relates to muscle recruitment during mobility tasks as shown by significant correlations of extent and magnitude of cortical activity with a number of EMG channels active during lower limb tasks in bilateral CP [41], and greater bilateral EEG-EMG coherence in the gamma-band with the hallucis longus in unilateral CP compared to TD during single stance [39].

These findings suggest that neural damage during brain development causes regional increases in cortical activation during a variety of mobility tasks in CP, and these are associated with greater task difficulties, poorer selective voluntary motor control, and increased co-activation of muscles. The brain injury is the precipitating event that alters early motor capabilities that in turn influence brain development and organization which further affect motor output. As an example, if an infant is moving abnormally due to muscle weakness or imbalance, even in the absence of a brain injury, the brain will develop abnormally [52]. Efforts are now underway to detect CP sooner and intervene earlier in this process to favor brain organization and produce a more functional motor trajectory [53,54].

Brain reorganization patterns are variable and likely to impact mobility

A relevant factor towards understanding the functional organization of the brain is the patterns of cortical representation and their potential reorganization after early injuries, which result from competitive processes early in development [10]. Nevertheless, although this aspect has been frequently investigated in studies of upper limb motor function [50], their relations with mobility remain less well understood. In this review, results on the spatial organization of the brain areas active during the recruitment of lower limb muscles and mobility tasks failed to show a consistent pattern.

Kesar et al. (2012) focused on TMS mapping of hand (first dorsal interosseus) representation, as activity in the lower limb muscles (tibialis anterior) was rarely elicited, possibly due to challenges in stimulating muscles controlled by deeper areas of the cortex, thus limiting the understanding of their cortical representation [33]. The findings were predominantly ipsilateral representation of the hand in those with unilateral CP (this subgroup also had participants with contralateral and bilateral patterns), and predominantly contralateral in those with bilateral CP (some participants with bilateral representation). In a similar way, Sukal-Moulton et al. (2018) reported more frequent contralateral activation during unilateral dorsiflexion and hip flexion during cycling tasks for the CP group, highlighting the variability that may result from particular injury and/or neuroplasticity events [41].

Another finding was the lateral shift in the activation in bilateral CP compared to TD individuals [41] and in the location of the maps eliciting lower limb muscle activity [29]. This finding may reflect reorganization related to the etiology of the brain injury (e.g., enlarged ventricles), and this plus a spread of activation related to synergistic muscle activation potentially results in an overlap in the representation of different areas, as suggested by Kesar et al. when reporting the proximity of first dorsal interosseus and tibialis anterior maps [33]. Weinstein et al. (2018) investigated the relative feasibility, reliability, and comparative results of TMS, EEG, and MRI in CP and concluded that each child with CP has a unique pattern of brain organization which is reflected in the heterogeneity that has been uncovered to date [55]. They further showed that different imaging modalities may yield different results as to which side(s) of the brain are activated during the same task in the same child, which makes interpretation challenging. This also supports the need for more individualized brain assessment in prescribing intervention strategies in CP since a specific pattern of brain organization cannot yet be assumed based on the motor performance of the CP subtype alone.

Microstructural organization of the corticospinal tract has an unclear role in mobility

The integrity and functioning of corticospinal projections have been acknowledged as important predictors of motor function after brain injuries [11]. This was confirmed by Grecco et al. (2016) who demonstrated that individuals with CP with the presence of motor-evoked potentials (MEPs) (quadriceps muscles) showed higher gait speeds than those with absent MEPs. In their study, this factor was suggested to be a potential predictor of responsiveness to brain stimulation interventions and is an aspect to be further explored [31].

Another method to explore the microstructural organization of the corticospinal tract (CST) is through DTI, a technique that quantifies the integrity of the white matter tract by the movement of water molecules [49]. Of the five studies using DTI, two focused on the posterior limb of the internal capsule (PLIC) [37,38], where descending motor axons that mediate voluntary motor control are more densely concentrated, and one addressed the posterior thalamic radiations [32]. Azizi et al. (2021) found a positive correlation between walking speed and fractional anisotropy (FA) and a negative correlation between Timed Up and Go test scores and FA of the more affected side of the brain [27]. However, the sample had to be divided according to their CST appearance, due to high inter-subject variability and these results did not remain consistent [27]. Corroborating with these findings, Damiano et al. (2022) found no significant correlations between DTI features and functional measures, only an association between higher precentral gyrus–midbrain connectivity and better mobility [28]. Similarly, in other studies, no clear associations were found between the severity of the CST injury with ambulation status [32] and between the FA and diffusivity of the PLIC with gait scores [37]. In contrast, Rose et al. (2007) suggested that the level of motor severity may be predicted by measures of integrity of the PLIC in participants with CP [38]. This study, however, had fewer participants and a relatively more heterogeneous sample compared to the others [38].

The lack of details on how gait was assessed in those with greater involvement limits the understanding of the origin of the differences. Again, the heterogeneity in the timing, type, and severity of CP across studies, as well as individual differences across even highly similar children, suggests the need for larger (combined) datasets that may enable us to better categorize children into meaningful sub-groups [28].

Several other factors may contribute to the lack of clear brain–behavior associations, such as the choice of tasks, a higher inter-subject variability [27] and the possible role of other areas of the brain in mobility. One example of these areas was suggested by Hoon et al. (2009), who reported a significant correlation between the severity of posterior thalamic radiation and ambulation deficits, highlighting the potential role of sensory and motor interactions in explaining motor outcomes [32]. This is supported by the MEG study by Kurz et al. (2015), where children with lower activity in the somatosensory cortex after peripheral tactile stimulation presented with lower gait speed [35]. Doucet et al. (2021) demonstrated reduced functional connectivity between sensorimotor, visual, and auditory networks in young people with CP when compared with their matched controls, which was shown to be associated with reduced step length [29]. These findings suggest abnormal functional integration of the brain’s motor and primary sensory systems in this population [29] and warrant further investigation.

Brain mapping techniques may be able to predict mobility outcomes

An increased somatosensory cortical activity at rest predicted increased somatosensory cortical activity during movement, and increased somatosensory cortical activity during movement predicted faster walking speed and increased step length in children with diplegic CP [40]. Similarly, FA of the more affected CST predicted 40% of the variability in walking speed and 38% in TUG in children with unilateral and bilateral CP [27]. In light of these findings, functional mapping may be an interesting resource for long-term outcome prediction, which can guide the choice of individualized interventions.

Pagnozzi et al. (2016) demonstrated that altered cortical structural shape characteristics (i.e., cortical thickness, curvature, and sulcal depth) in children with unilateral CP can predict multiple outcomes, such as communication, visual, and upper limb function [56]. Indeed, structural brain mapping methods have been widely used to predict motor type and topography in infants with CP [53,57,58,59].

### 4.4. Methodological Quality of the Included Studies

Six out of the fifteen studies included had good methodological quality. The remaining nine studies presented moderate limitations.

Studies failed to report and justify the sample size method, which is considered an important component of the methodological quality. Few studies specified the study design, providing broad information (e.g., “in this exploratory investigation”). Studies also failed to provide information about the recruitment method, which may limit the replication of studies [26].

### 4.5. Clinical and Research Considerations

Taking the results together, children with CP may have altered brain activity during mobility tasks when compared with TD and across various mobility outcomes. These findings highlight the neural basis of limitations in activities that are important for individuals with CP and might reinforce the importance of implementing motor intervention strategies that explore principles of neuroplasticity [16] to minimize maladaptive changes in brain activity during mobility activities and maximize the recovery potential of children with CP. Examples of these therapies include Hand and Arm Bimanual Intensive Therapy Including Lower Extremities (HABIT-ILE) [60] and Constraint-Induced Movement Therapy (CIMT), as they have been suggested to potentially change brain activity [61]. It is important to note that successful interventions also demonstrate a range of effectiveness across participants, which may be related in part to their variable patterns of brain organization that may be positively or potentially negatively affected by specific approaches depending on which patterns are more adaptive for each individual.

Limited information on the associations of brain activity with mobility can lead to difficulties in understanding the neurodevelopmental and neuroplasticity processes in the face of interventions focused on mobility tasks. In this review, we aimed to address this gap, despite the challenges in standardizing results due to the high variability in characteristics of participants with CP and assessment instruments. Our goal was also to encourage further research on this topic.

## 5. Conclusions

Children with CP reveal increased brain activation during mobility compared to TD children. The patterns of brain reorganization and CST microstructural organization of children with CP during mobility activities showed high variability, perhaps due to the heterogeneity of participants and methods. Further studies are necessary to understand the relationship between brain, behavior, and mobility tasks and thus, to facilitate the translation of findings to clinical practice. Such information could guide neurorehabilitation strategies targeting brain neuroplasticity accompanying functional gains.

## Figures and Tables

**Figure 1 ijerph-21-01039-f001:**
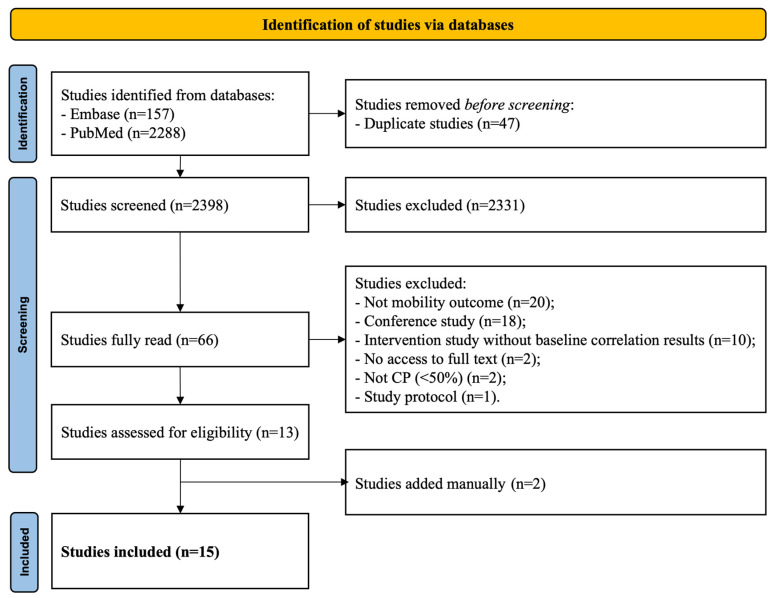
Study flowchart.

**Table 1 ijerph-21-01039-t001:** Characteristics of the study samples.

Author, Year	Study Design	Sample Size (*n*)	Sex	Age (Mean ± SD)	GMFCS Levels	CP Type
Azizi et al., 2021 [27]	Observational	52 CP: 26 CP group 1: 14 CP group 2: 12 TD: 26	CP: 14 (F) and 12 (M) TD: 12 (F) and 14 (M)	CP: 9.03 ± 2.73 y TD: 10.77 ± 3.03 y	I, II, III, IV	Diplegic; Hemiplegic
Damiano et al., 2022 [28] *	Observational	34 CP: 16 TD: 18	CP: 13 (F) and 3 (M) TD: 10 (F) and 8 (M)	CP: 13.3 ± 3.3 y TD: 11.3 ± 3.4 y	I, II, III	Bilateral
Doucet et al., 2021 [29] *	Observational	65 CP: 27 TD: 38	CP: 12 (F) and 15 (M) TD: 14 (F) and 24 (M)	CP: 16.93 ± 4.90 y TD: 14.44 ± 2.35 y	I, II, III, IV	Diplegic; Hemiplegic
George et al., 2020 [30]	Observational	20 CP: 10 TD: 10	CP: 8 (F) and 2 (M) TD: 8 (F) and 2 (M)	CP: 15.1 ± 3.9 y TD: 15.0 ± 3.2 y	I, II	Unilateral
Grecco et al., 2016 [31]	Intervention	CP: 56	-	8.2 ± 1.6 y	I, II, III	Diplegic; Hemiplegic
Hoon JR et al., 2009 [32]	Observational	63 CP: 28 TD: 35	CP: 12 (F) and 16 (M) DT: 16 (F) and 19 (M)	CP: 5 y 10 m ± 2 y 6 m TD: 5 y 9 m ± 4 y 4 m	-	Ataxic; Diplegic; Hemiplegic; Quadriplegic
Kesar et al., 2012 [33]	Observational	CP: 13 Hemiplegic: 8 Diplegic: 6	9 (F) and 4 (M)	12.08 ± 2.3 y	I, II	Diplegic; Hemiplegic
Kurz et al., 2014 [34]	Observational	12 CP: 4 TD: 8	CP: 2 (F) and 2 (M) TD: 7 (F) and 1 (M)	CP: 11.0 ± 4 y TD: 13.2 ± 3 y	II, III	Diplegic
Kurz et al., 2015 [35]	Observational	22 CP: 11 TD: 11	-	CP: 14.5 ± 0.7 y TD: 14.1 ± 0.7 y	I, II, III	Diplegic; Hemiplegic
Kurz et al., 2020 [36]	Observational	35 CP: 20 TD: 15	CP: 8 (F) and 12 (M) TD: 5 (F) and 10 (M)	CP: 15.5 ± 3 y TD: 14.1 ± 3 y	I, II, III	Diplegic
Meyns et al., 2016 [37] ^†^	Observational	CP: 50	18 (F) and 32 (M)	Unilateral CP: 6 y 10 m ± 2 y 1 m Bilateral CP: 6 y 7 m ± 2 y 1 m	I, II, III	Unilateral; Bilateral
Rose et al., 2007 [38] ^†^	Observational	24 Normal FA: 14 Low FA: 10 (7 CP)	Normal FA: 7 (F) and 7 (M) Low FA: 4 (F) and 6 (M)	Normal FA: 4.4 ± 0.2 y Low FA: 4.2 ± 0.4 y	I, II, V	Diplegic; Hemiplegic; Quadriplegic
Short et al., 2020 [39]	Observational	21 CP: 9 TD: 12	CP: 7 (F) and 2 (M) TD: 8 (F) and 4 (M)	CP: 16.0 ± 2.7 y TD: 14.8 ± 3.0 y	I, II	Unilateral
Trevarrow et al., 2022 [40]	Observational	47 CP: 22 TD: 25	CP: 10 (F) and 12 (M) TD: 6 (F) and 19 (M)	CP: 14.54 ± 0.91 y TD: 14.50 ± 0.53 y	I, II, III, IV	Diplegic
Sukal-Moulton et al., 2018 [41]	Observational	28 CP: 14 TD: 14	CP: 9 (F) and 5 (M) TD: 5 (F) and 9 (M)	CP: 17.6 ± 9.6 y TD: 17.2 ± 9.6 y	I, II, III	Bilateral

Legend: CP: cerebral palsy; TD: typical development; F: female; M: male; FA: fractional anisotropy; y: year; m: month). * Participant demographic characteristics at baseline assessment. ^†^ Sample characterization by the time of mobility assessment.

**Table 2 ijerph-21-01039-t002:** Main results of cerebral and mobility outcomes.

Brain Mapping Technology	Author, Year	Device Parameters	Brain Region and Related Outcomes	Mobility Measure and Outcomes	Associations of Brain Activity with Mobility *
EEG	George et al., 2020 [30]	- 64-channel; - 10–20 system.	- C3–C4 (sensorimotor cortex); - Coherence of central and frontal hemisphere; central and parietal hemisphere; - Peak mu band frequency, mu desynchronization;	- Quiet standing; - Treadmill walking at self-selected speed for 5 minutes; Gait speed.	- Bilateral desynchronization over motor cortex during walking in CP and TD, with lower peaks in CP; - Greater power in standing than treadmill walking in both hemispheres in TD and non-dominant hemisphere for CP (n.s. trend in dominant); - Greater central-frontal coherence in CP than TD in non-dominant hemisphere during quiet standing; - Greater coherence in TD than CP in dominant hemisphere during quiet standing.
	Short et al., 2020 [39]	- 64-channel; - 10–20 system; - Frequency 1000 Hz.	- Frontal, dominant and non-dominant parietal, and motor, and non-dominant prefrontal regions; - Frequencies: 8 to 13 Hz (mu-band) 14 to 30 Hz (beta-band);	- Comfortable walking speed (kinematic gait analysis) Gait speed; Cadence; Stance time; Step length. - EMG recorded wirelessly during gait from bilateral the tibialis anterior, medial gastrocnemius, soleus, peroneus longus, rectus femoris, vastus lateralis, medial hamstrings, and hallucis longus.	- Mu-band ERD in motor clusters during gait in CP and TD; - No sustained beta-band ERD in TD; - CP group: Greater cortical activation during walking (mu- and beta-ERD in the dominant and non-dominant motor and parietal regions; elevated low gamma-activity in frontal, parietal and non-dominant motor areas) than TD; Greater bilateral motor EEG-EMG coherence in gamma-band for hallucis longus than TD; - Six cortical clusters identified as having gait-related activation and all contained participants from both CP and TD groups; - Non-dominant motor cluster least represented in CP; - Greater mu-suppression in TD than CP in dominant motor cluster.
fNIRS	Kurz et al., 2014 [34]	- Continuous wave fNIRS system; - 695 and 830 nm; - Sampled at 10 Hz; - 8 infrared optode emitters and 8 detectors; - 24-channels; - 10/20 system.	- Supplementary motor area, precentral gyrus, postcentral gyrus, and superior parietal lobule.	- Gait: 30 s of walking at a speed of 0.45 m/s on a programmable treadmill; Stride, stance, and swing times; Coefficient of variation of temporal kinematic measures. - Five alternating blocks of standing still for 30 s.	- Greater [HbO] in precentral gyrus, postcentral gyrus, and superior parietal lobule in CP than TD during walking; - Strong positive correlations between [HbO] in superior parietal lobule and variability in stride and stance time intervals for all participants; - Negative correlation between variability in the stride time intervals and the [HbO] in postcentral gyrus; - Positive correlation between stance time intervals and the [HbO] in pre- and post-central gyri; - These correlations relate to gait errors.
	Sukal-Moulton et al., 2018 [41]	- Continuous-wave CW6 NIRS system; - 690 nm and 830 nm wavelengths; - 8 sources and 16 detectors with inter-optode distance 22.4–36.5 mm; - Data collection at 50 Hz; - 10/20 System.	- Bilateral sensorimotor areas; - Extent of activation (number of channels active); - Magnitude of activation (sum of beta values).	- Motor tasks: Single leg cycling; Bilateral cycling. - Clinical scales: GMFCS; PEDI-CAT; AbiLOCO.	- Group effect on extent of activation: For unilateral cycling, GMFCS level III had more active channels than TD. - Group effect on magnitude for all tasks except unilateral hip flexion: Higher values in GMFCS level III than all other groups except bilateral dorsiflexion where they were only higher than TD. - Positive correlations of extent and magnitude of activity with number of EMG channels active during cycling; - Contralateral hemisphere more active for GMFCS level III during unilateral and bilateral cycling compared to ipsilateral; - Ipsilateral hemisphere more active for GMFCS I during unilateral cycling; - Activation for CP more lateral in the contralateral hemisphere than in TD.
MEG	Kurz et al., 2015 [35]	- 306-sensor Elekta MEG system; - 4 to 14 Hz time- frequency range.	- Amplitude of the peak latency.	- Walking at preferred and fast-as-possible speed using gait mat: Gait velocity; Step length; Cadence.	- All participants: Preferred walking speed: positive correlation between step length and negative correlation between cadence with amplitude of the peak voxel; Fast-as-possible speed: positive correlation between amplitude of peak voxel and step length; - CP participants: Preferred and fast-as-possible speed positively correlated with amplitude of the peak voxel. Fast-as-possible speed: Step length positively correlated with amplitude of same peak voxel.
MEG + MRI	Kurz et al., 2020 [36]	○ MEG: - 306-sensor Elekta MEG system; - Bandwidth of 0.1–330 Hz. ○ MRI: - Siemens Prisma 3T scanner; - 64-channel head/neck coil.	- Strength of the beta ERD; - MEG behavioral variables.	- Walking at preferred speed using gaitRITE digital mat Gait velocity Step length; Cadence.	- All participants: Positive rank-order correlation between the strength of the beta ERD during the motor planning and execution stage and the participant’s walking speed and cadence. - CP participants: Positive rank-order correlation between the strength of the beta ERD during the motor execution stage and the preferred walking speed and cadence.
	Trevarrow et al., 2022 [40]	○ MEG: - 306-sensor Elekta MEG system; - Bandwidth of 0.1–330 Hz. ○ MRI: - Skyra 3T scanner; - 32-channel head coil.	- Somatosensory cortex; - Peak voxel coordinates; - Magnitude of the somatosensory cortical response, and neural time courses.	- Walking at as-fast-as-possible speed on gaitRITE digital mat. Gait velocity; Step length; Cadence.	- Indirect effect on the relationship between passive somatosensory cortical activity and both walking velocity and step length; - Increased somatosensory cortical activity at rest predicted increased somatosensory cortical activity during ankle plantarflexion; - Increased somatosensory cortical activity during movement predicted faster walking velocity and increased step length.
fMRI + DTI;	Damiano et al., 2022 [28]	○ fMRI: - Philips 3.0T Achieva scanner; - T1, T2, FLAIR, resting state BOLD-fMRI, MPRAGE. ○ DTI: - Single-shot EPI with SENSE acquisition; - Echo time 66 ms.	- Resting-state functional connectivity; - FA.	- Motor task: Free and fast gait speed Free and fast elliptical - Clinical scales: GMFCS PEDI-CAT	- Higher precentral gyrus–midbrain connectivity associated with better mobility.
	Hoon et al., 2009 [32]	○ MRI: - 1.5T scanner; - Sagittal and axial T1/T2 sequences. ○ DTI: - Repetition time 6.2 s and 9.4 s; - Echo time 80 ms; - Maximum b value was 700 s mm^2^, in a scheme of 30 different gradient directions along with five reference images with minimal diffusion weighting.	○ MRI: - Thalamocortical pathways connected to the sensory cortex and descending corticospinal tracts. ○ DTI: - FA; - Vector maps; - Color-coded maps; - Ordinal scale for grading white matter tracts.	- Ambulatory status or best motor skill attained	- Ambulation deficits positively associated with severity of posterior thalamic radiation injury, but not with corticospinal tract injury.
	Meyns et al., 2016 * [37]	○ MRI: - 3-T system and 1.5-T system; - 8-element sense head coil. ○ DTI: - 3-T scanner; - Single-shot spin echo.	○ DTI: - CST defined between the primary motor cortex and midbrain; - CST_PLIC defined as segment that runs through the posterior limb of the internal capsule; - Asymmetry index of FA and apparent diffusion coefficients.	- Walking on a 10 m walkway at self-selected speed. Spatiotemporal parameters; Joint angles; Internal moments; Powers normalized to body mass. - Gait Profile Score	- No correlation between mobility measures and DTI variables.
	Rose et al., 2007 [38]	○ MRI: - 1.5T system; - T1, T2, FLAIR, and gradient echo sequences. ○ DTI: - Single-shot echo planar technique; - Diffusion measured along six non-collinear directions.	- PLIC; - FA;	- Walking in straight 5 m walkway at a self-selected speed; Gait NI.	- Low neonatal FA: Strong negative correlation between FA of combined right and left PLIC and log NI at 4 years; GMFCS higher for children with low FA; Negative correlation between FA of combined right and left PLIC and GMFCS values at 4 years; Strong positive correlation between GMFCS and log NI values.
rs-fMRI	Doucet et al., 2021 [29]	- 3-T Siemens Skyra MRI scanner; - 32-channel head coil; - T2 sequence; - Single-shot echo planar gradient echo imaging sequence.	- Left precentral cortex; left lingual gyrus; left Heschl gyrus - Resting-state networks	- Walking at preferred speed on gaitRITE digital mat. Velocity; Step length and width; Cadence.	- Higher functional connectivity between the left precentral and lingual gyri associated larger right step length.
DTI + TMS	Azizi et al., 2021 [27]	○ DTI: - 3-T scanner with sedation; - T1-, T2-weighted FLAIR, and DTI images; - Repetition time/echo time of 9100/90 ms. ○ TMS: - Magstim Rapid^2^ stimulator; - Single-pulse TMS with a 70% and then an increase of 10%.	- Corticospinal tract ○ DTI: - FA; - MD; - AD; - RD. ○ TMS: - Resting motor threshold - MEP peak-to-peak amplitude - MEP latency	- 10-meter Walk Test; - TUG; - 6-minute Walk Test.	- All participants: Walking speed negatively correlated with MD and RD of the more affected side of the brain; Walking speed positively correlated with FA of the more affected side of the brain; TUG negatively correlated with FA and positively correlated with MD and RD of the more affected side of the brain. - CP group 1: Negative correlation between speed and RMT of the more affected side; Positive correlation between RMT and TUG of both hemispheres; FA of the more affected CST significantly predicted 40% of the walking speed and 38% (R2 = 0.381) of TUG performance. - CP group 2: No significant correlation
TMS	Grecco et al., 2016 [31]	- Set to 110% of resting motor threshold.	- 1 cm anterior to 3 cm posterior to the vertex; 2 cm over the left and right motor cortices. - MEP registered from the quadriceps muscle;	- 6-minute Walk Test; - GMFM-88 dimension E; - Gait speed (3d motion); - Gait Profile Score.	- At baseline: Children with present MEP had higher gait speed;
	Kesar et al., 2012 [33]	- MagStim 200 stimulator with a double 7 cm circular coil; - TMS intensity at 120% motor threshold EEG, cap centered on vertex, and oriented in antero-posterior and medio-lateral directions.	- MEPs recorded using surface EMG sensors attached to bilateral tibialis anterior and first dorsal interosseous muscles.	- GMFM-66	- Unilateral CP: Affected (right) hand (first dorsal interosseus) motor cortical representation was ipsilateral (*n* = 4), contralateral (*n* = 2), or bilateral (*n* = 1). - Bilateral CP: Bilateral and contralateral maps related to higher Melbourne and GMFM scores; Tibialis anterior map data obtained from 4/13 (three hemiplegia, one diplegia); tibialis anterior MEPs not elicited in the others.

Legend: EEG: electroencephalography; CP: cerebral palsy; TD: typical development; Hz: hertz; ERD: event-related desynchronization; EMG: electromyography; fNIRS: functional near infrared spectroscopy; nm: nanometer; [HbO]: oxygenated hemoglobin concentration; mm: millimeter; GMFCS: Gross Motor Function Classification System; PEDI-CAT: Pediatric Evaluation of Disability Inventory–Computer Adaptive Test; AbiLOCO: Locomotor Ability Measure; MEG: magnetoencephalography; ms: milliseconds; MRI: magnetic resonance imaging; fMRI: functional magnetic resonance imaging; DTI: diffusion tensor imaging; s: seconds; CST: corticospinal tract; FA: fractional anisotropy; PLIC: posterior limbs of the internal capsule; NI: normalcy index; TMS: transcranial magnetic stimulation; MD: mean diffusivity; AD: axial diffusivity; RD: radial diffusivity; MEP: motor-evoked potential; cm: centimeters; RMT: resting motor threshold; TUG: Timed Up and Go test; GMFM: gross motor function measure. * Significant associations reported.

**Table 3 ijerph-21-01039-t003:** Quality assessment of the reviewed studies.

Subitens	1	2	3	4	5	6	7	8	9	10	11	12	13	14	15	16	Total Score (0–16)	Quality
Azizi et al., 2021 [27]	1	0	0	1	1	0	0	1	0	0	1	1	1	1	1	1	10	Fair
Damiano et al., 2022 [28]	1	1	1	1	1	0	0	1	1	0	1	1	1	1	1	1	13	Good
Doucet et al., 2021 [29]	1	1	0	1	1	0	0	1	1	0	1	1	1	1	1	1	12	Good
George et al., 2020 [30]	1	1	0	1	1	0	0	1	1	0	1	1	1	1	1	1	12	Good
Grecco et al., 2016 [31]	1	1	1	1	1	0	0	1	0	0	1	1	1	1	1	1	12	Good
Hoon JR et al., 2009 [32]	1	1	1	1	1	1	0	1	1	0	1	1	1	1	1	1	14	Good
Kesar et al., 2012 [33]	1	1	1	1	1	1	0	1	1	0	1	1	1	1	1	1	14	Good
Kurz et al., 2014 [34]	1	1	1	1	0	1	0	1	0	0	1	1	1	1	1	0	11	Fair
Kurz et al., 2015 [35]	1	0	0	1	0	0	0	1	0	0	1	1	1	1	1	0	8	Fair
Kurz et al., 2020 [36]	1	1	0	1	1	0	0	1	0	0	1	1	1	1	1	1	11	Fair
Meyns et al., 2016 [37]	1	0	0	1	1	1	0	1	1	0	1	1	0	1	1	1	11	Fair
Rose et al., 2007 [38]	1	0	0	1	0	0	0	1	0	0	1	1	1	1	1	0	8	Fair
Short et al., 2020 [39]	1	1	0	1	0	0	0	1	1	0	1	1	1	1	1	1	11	Fair
Trevarrow et al., 2022 [40]	1	1	0	1	1	0	0	1	0	0	1	1	1	1	1	0	10	Fair
Sukal-Moulton et al., 2018 [41]	1	1	0	1	1	0	0	1	0	0	1	1	1	1	1	1	11	Fair
% of studies that scored each item	100	73	33	100	73	33	0	100	53	0	100	100	93	100	100	73	-	-

Legend: (1) positive; (0) negative.

## Data Availability

The raw data supporting the conclusions of this article will be made available by the authors on request.

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
