# Peer review of "Neural Correlates of Mobility in Children with Cerebral Palsy: A Systematic Review"

_ijerph, 2024, doi:10.3390/ijerph21081039_

Round 1

Reviewer 1 Report

Comments and Suggestions for Authors

·      To enhance the reporting of the abstract, the authors are encouraged to follow the PRISMA 2020 Abstract checklist.

·      The introduction has done well to introduce the topic (i.e., CP and neural correlates). However, it is not clear on the need to systematically review and synthesize evidence. Has there been previous attempts to systematically review the topic? What is the current state of synthesized evidence on the topic? How different is the current review from previous reviews?

·      I refer back the authors to the PRISMA 2020 statement. There are items that have been deemed N/A. Certain information on the methods are not properly reported. These items should be carefully reviewed in an attempt to succinctly report the minimum information that should be reported in systematic reviews.

·       The limited information sources searched in this review raises concerns on whether attempts to fully search all relevant literature has been performed in order to avoid bias.

·      CP is typically associated with other conditions. The authors excluded “health conditions.” This is unclear and should be explained.

·      The authors responsible for the searching, appraisal, data extraction, and synthesis should e reported.

·      What was the rationale for the n=2 studies that were added manually. How were these searched? Why were these not captured in the systematic searching?

·      Were there studies whose population included participants beyond the upper age limit? How did you ensure that the extracted data for specifically for the age range concerned in this review?

·      Given the outcome of association between neurophysiological correlates of functional mobility, why was a meta-analysis not possible?

·      It might be useful to explicitly include the experimental paradigm.

·      The data analysis mentioned that frequency analysis and averaged data were to be performed. However, this is not apparent in the results and/or discussion reporting.

·      I find the section on Clinical and Research Considerations quite useful. However, the authors will need to reflect on the limitations and contexts of their review to avoid overgeneralization. The lack of methods to ensure certainty in the recommendations raises additional concern on the believability of these implications.

Author Response

  •     To enhance the reporting of the abstract, the authors are encouraged to follow the PRISMA 2020 Abstract checklist.

RESPONSE: Thank you for your suggestion. We made some additions according to the PRISMA abstract checklist. 

  •     The introduction has done well to introduce the topic (i.e., CP and neural correlates). However, it is not clear on the need to systematically review and synthesize evidence. Has there been previous attempts to systematically review the topic? What is the current state of synthesized evidence on the topic? How different is the current review from previous reviews?

RESPONSE: We appreciate your suggestion and have listed previous reviews, which focus on structural imaging as opposed to our choice to focus on functional mapping approaches.

  •     I refer back the authors to the PRISMA 2020 statement. There are items that have been deemed N/A. Certain information on the methods are not properly reported. These items should be carefully reviewed in an attempt to succinctly report the minimum information that should be reported in systematic reviews.

RESPONSE: Thank you for pointing this out. Improvements were made. 

  •       The limited information sources searched in this review raises concerns on whether attempts to fully search all relevant literature has been performed in order to avoid bias.

RESPONSE: We appreciate your point. We highlight that, according to AMSTAR-2 (A MeaSurement Tool to Assess systematic Reviews), a comprehensive literature search in at least two databases is sufficient to incorporate the main studies on a topic and to score as a high-quality review. It’s important to highlight that we searched the two primary databases in our field. Additionally, a manual search was conducted to enhance our literature review. We are therefore confident that we were able to retrieve most available studies and that the limited findings actually reflect literature gaps.

  •     CP is typically associated with other conditions. The authors excluded “health conditions.” This is unclear and should be explained.

RESPONSE: We agree with you that CP is typically associated with other comorbidities. However, in the sentence “Studies were excluded if the participants had other health conditions than CP” we were referring to other health conditions not related to a CP diagnosis such as traumatic brain injury, genetic disorders, and others. We added this information in the main text. 

  •     The authors responsible for the searching, appraisal, data extraction, and synthesis should e reported.

RESPONSE: Thank you for your suggestion. The information was added. 

  •     What was the rationale for the n=2 studies that were added manually. How were these searched? Why were these not captured in the systematic searching?

RESPONSE: As mentioned above, a manual search is an essential step in systematic reviews, according to AMSTAR-2. After completing the selection from PubMed and Embase, a manual search is recommended to include studies that were not detected in the initial search.

  •     Were there studies whose population included participants beyond the upper age limit? How did you ensure that the extracted data for specifically for the age range concerned in this review?

RESPONSE: We highlight that, as described in the inclusion criteria, studies in which the mean participant age were up to 18 years were included in our systematic review. All included studies met this criteria and for further information on individual studies, detailed information regarding the sample characteristics, including age can be found in Table 1. 

  •     Given the outcome of association between neurophysiological correlates of functional mobility, why was a meta-analysis not possible?

RESPONSE: Meta-analysis was not possible due to the characteristics and results of the studies. Despite being categorized under "mobility," the outcomes were too heterogeneous, making it difficult to conduct a meta-analysis (doi: 10.1186/0778-7367-71-21 ;  https://doi.org/10.1136/bmj.d7762 )

  •   It might be useful to explicitly include the experimental paradigm.

RESPONSE: Thank you for your suggestion. The information was added in Table 1. 

  •     The data analysis mentioned that frequency analysis and averaged data were to be performed. However, this is not apparent in the results and/or discussion reporting.

RESPONSE: We appreciate you for noticing this mistake. Changes were made in the data analysis description. 

  •     I find the section on Clinical and Research Considerations quite useful. However, the authors will need to reflect on the limitations and contexts of their review to avoid overgeneralization. The lack of methods to ensure certainty in the recommendations raises additional concern on the believability of these implications.

RESPONSE: We appreciate your suggestion and state that changes were made to moderate our statements.

Reviewer 2 Report

Comments and Suggestions for Authors

This systematic review effectively summarizes research on brain activity associated with mobility of children with cerebral palsy. The manuscript is well written. The search strategy, selection criteria, methodological quality of the studies reviewed, and results are clearly described and meet the criteria for a good systematic review. The Tables summarize the characteristics of the children included in each study, how and where brain activity was measured, how mobility was measured, and findings. The results and discussion are nicely organized and carefully worded to communicate the wide variability in findings and many knowledge gaps.      

I have two recommendations to improve understanding. First is to briefly define terms used to describe brain activity such as alpha (mu), beta, and gamma bands, task related desynchronization, and central-frontal coherence. This could be in a subsection proceeding the Results. Second, since recoding brain activity while a child is performing a mobility task represents the ideal situation, I recommend summarizing the results of the four studies were this occurred, and any insights compared to the studies were brain activity was not recorded during movement.

Author Response

This systematic review effectively summarizes research on brain activity associated with mobility of children with cerebral palsy. The manuscript is well written. The search strategy, selection criteria, methodological quality of the studies reviewed, and results are clearly described and meet the criteria for a good systematic review. The Tables summarize the characteristics of the children included in each study, how and where brain activity was measured, how mobility was measured, and findings.

The results and discussion are nicely organized and carefully worded to communicate the wide variability in findings and many knowledge gaps. I have two recommendations to improve understanding. First is to briefly define terms used to describe brain activity such as alpha (mu), beta, and gamma bands, task related desynchronization, and central-frontal coherence. This could be in a subsection proceeding the Results. 

RESPONSE: We appreciate your suggestion and have briefly described this in the introduction section, aligning it with the reasoning presented in that section.

Second, since recoding brain activity while a child is performing a mobility task represents the ideal situation, I recommend summarizing the results of the four studies were this occurred, and any insights compared to the studies were brain activity was not recorded during movement.

RESPONSE: Thank you for your suggestion. Besides what had been included in the discussion session, item 4.3, we added some more information in results section, item 3.3, the last paragraph, for further clarification. 

Reviewer 3 Report

Comments and Suggestions for Authors

Review

In general, is an interesting paper that highlight important issues, despite studies variability. The structure is good following most PRISMA recommendations, however come points should be clarified.

Introduction

When describe functional mobility tasks authors just mention gait parameters. Please cite others examples and explaining why tasks as quiet standing was considered in the present study.

Methods

Search Strategy: when research begin?

Study Selection: Are there any time restrictions for study inclusion?

Methodological quality assessment: Better explain the tool used to bias evaluation as number of criteria and main characteristics of these criteria.

Results

According Functional Mobility definition cited in the introduction, some outcomes cited in Table 2 should not be included as Quiet standing and Motor tasks: Ankle dorsiflexion and Hip flexion.  Please better clarify why this task was considered as a functional mobility activity.

Discussion

Some information cited in Discussion should be added in results section, as line 196 and 204, Line 214 and 215

Conclusion

Line 405 “perhaps due to the heterogeneity of participants and methods” this information should be in discussion section.

Table 2 layout should be better structured to reader understanding. For example, icons can be standardized.

Author Response

In general, is an interesting paper that highlight important issues, despite studies variability. The structure is good following most PRISMA recommendations, however come points should be clarified.

Introduction

When describe functional mobility tasks authors just mention gait parameters. Please cite others examples and explaining why tasks as quiet standing was considered in the present study.

RESPONSE: Thank you for your suggestion. We used the International Classification of Functioning, Disability and Health ( ICF) as the primary reference to define mobility. In Chapter 4 of the Activity and Participation section, the ICF outlines the key activities related to this category. We have added this information to the main text and changed the terminology from "functional mobility" to simply "mobility" to avoid any confusion.

Methods

Search Strategy: when research begin?

RESPONSE: The database search began in early April and was completed within the same month. We updated the terminology in the main text for better clarity.

Study Selection: Are there any time restrictions for study inclusion?

RESPONSE: No, there weren't. We included all relevant studies retrieved in the search, with no time restrictions. 

Methodological quality assessment: Better explain the tool used to bias evaluation as number of criteria and main characteristics of these criteria.

RESPONSE: Thank you for your suggestion. Improvements were made in the main text. 

Results

According Functional Mobility definition cited in the introduction, some outcomes cited in Table 2 should not be included as Quiet standing and Motor tasks: Ankle dorsiflexion and Hip flexion.  Please better clarify why this task was considered as a functional mobility activity.

RESPONSE: Thank you for pointing this out. Quiet standing was considered mobility, as it is included in ICF chapter 4 (as mentioned above). However, we state that information regarding joint movements was excluded from Table 2.

Discussion

Some information cited in Discussion should be added in results section, as line 196 and 204, Line 214 and 215

RESPONSE: We appreciate your suggestion. However, to avoid duplicating content, we decided not to include the information in the Results texts, as the same content is reported in Table 2.

Conclusion

Line 405 “perhaps due to the heterogeneity of participants and methods” this information should be in discussion section.

RESPONSE: Thank you for your suggestion. Besides what had been included in the discussion session, we added some more information in item 4.5, the last paragraph, for further clarification. 

Table 2 layout should be better structured to reader understanding. For example, icons can be standardized.

RESPONSE: We appreciate your suggestion. Changes were made to improve the understanding.